# VISUAL ABSTRACT REASONING BY SELF-SUPERVISED CONTRASTIVE LEARNING

## ABSTRACT

Neuro-symbolic models of artificial intelligence (AI) have been recently developed to perform tasks involving abstract visual reasoning that is a hallmark of human intelligence but remains challenging for deep neural network methods. However, most of the current neuro-symbolic models also rely on supervised learning and auxiliary annotations, different from human cognitive processes that are much dependent on the general cognitive abilities of entity and rule recognitions, rather than learning how to solve the specific tasks from examples. In this work, we propose a neuro-symbolic model by self-supervised contrastive learning (NS-SSCL) with unique and invariant representations of entities and rules in the perception and reasoning modules, respectively, to solve Raven's Progressive Matrices (RPMs) and its variant, a typical type of visual reasoning task used to test human intelligence. The perception module parses each object into invariant representations of attributes. The reasoning module grounds the representations of object attributes to form the latent rule representations also through SSCL. Further, the relationships between the neural representations of object attributes and symbols used for rule reasoning are coherently mapped. Finally, the scene generation engine aggregates all attribute and rule representation distributions to produce a probabilistic representation of the target. NS-SSCL obtains state-of-the-art performance in unsupervised models to solve the RAVEN and V-PROM benchmarks, even better than most of the supervised models. The success of the proposed model suggests that construction of general cognitive abilities like humans may render the AI algorithms to solve complex tasks involving higher-level cognition such as abstract reasoning in a human-like manner.

## 1 INTRODUCTION

Abstract reasoning is essential for human intelligence. The capability of abstract reasoning in humans is domain-general and can be effectively estimated by a simple visual reasoning task test, such as Raven's Progressive Matrices (RPMs) (Raven et al., 1938). The premise of RPMs is that it does not rely on domain-specific knowledge or verbal ability, but the test performance is diagnostic of verbal, spatial and mathematical reasoning abilities (Carpenter et al., 1990; Snow et al., 1984). Hence, it is believed that solving RPMs by artificial intelligence (AI) might be a cornerstone toward artificial general intelligence (AGI) (Bilker et al., 2012; Zhang et al., 2019a). Although deep neural networks by supervised learning have achieved a great success in visual categorizing, such a neural network architecture is not versatile for visual reasoning (Hoshen & Werman, 2017; Barrett et al., 2018). The core feature in visual reasoning tasks (e.g., RPMs) is that the rules governing the organization of entities are semantically defined by the spatiotemporal relations between entities, but rather intrinsic to the entities per se (Lovett et al., 2007). Thereby, the semantics of entities and the underlying rules are weakly connected. This leads to end-to-end deep-learning (DL) algorithms are inefficiently to concurrently learn both properties. Although a number of variant DL models have been developed to achieve high performance superior to humans (Zhang et al., 2019b; Zhuo & Kankanhalli, 2020; Mańdziuk & Zychowski, 2019; Zhuo & Kankanhalli, 2021), these models are monolithic, lacking clear distinctions between the processes of perception and reasoning like in humans (Marcus & Davis, 2020; Fodor & Pylyshyn, 1988). To mimic the human-like processes involved in solving abstract reasoning tasks, some neuro-symbolic (NS) methods have been recently proposed to combine the deep neural network for the perception module with a reasoning module

for symbolic logic execution (Yi et al., 2019; Mao et al., 2019; Zhang et al., 2021). However, these models also need to learn the connections between the contexts of instances and the supervised answers from scratch as DL models.

In striking contrast, humans who have never previously met the problems can soon solve the abstract reasoning tasks. Humans do not rely on the task-specific experiences in solving the tasks, but their prior general cognitive abilities, namely, object and rule recognitions. Humans, even at a very early stage of life, can recognize tons of objects and their attributes (Spelke, 1990), and soon later recognize the rules governing the world and apply these rules to new situations (Gopnik et al., 2004). For these reasons, the RPM tests are capable of evaluating human's general reasoning abilities (Marcus & Davis, 2020; Fodor & Pylyshyn, 1988). Although it remains open questions about how humans learn and form the object and rule representations, a critical feature of these general cognitive abilities granted for abstract reasoning is that the attribute and rule representations are unique and invariant in the brain (Li & DiCarlo, 2008; Mansouri et al., 2020). For instance, we recognize the same color of 'green' from different objects and recognize the latent rule governing the color relationship across a set of objects. Thus far, it remains challenges to design an automatic AI algorithm behaving like humans in these tasks.

To better investigate abstract visual reasoning, RAVEN (Appendix A) and other RPM-like datasets have been proposed. In RAVEN dataset, each RPM problem consists of 9 panels in a form of $3 \times 3$ matrix with 8 context panels and a missing panel at the 9th entry (Matzen et al., 2010; Zhang et al., 2019a). The goal of the task is to find out the correct answer from 8 candidate panels that completes the matrix with satisfactions of the latent rules governing the organization of object attributes in the three continuous panels within each row and are consistent across the three rows in the matrix (Figure 1). Overall, the task requires two independent cognitive abilities of visual perception and rule reasoning. If visual perception module can perfectly recognize all object attributes, then the process of identifying the latent rules becomes plain and reduced to exhaustive search in the rule space (Matzen et al., 2010; Zhang et al., 2019a). Different from most of visual perception tasks, both the object attributes and rules are required to be identified. This is difficult for deep neural networks, and so far also remains challenges for the NS models. Additionally, the V-PROM task (Appendix F) is also an RPM-like task but with natural images (Teney et al., 2020). This new task increases the difficulty of visual perception.

Inspired by the above-mentioned human cognitive processes in abstract reasoning, we here demonstrate a human-like NS model can solve abstract visual reasoning in a human-like manner. To build up a non-verbal visual reasoning ability for an AI model on the basis of the general cognitive abilities in object and rule recognitions, we move a step further towards a NS model without supervisions of the answers or annotations, but with a self-supervised contrastive learning (SSCL) method to establish both the object and rule recognition abilities, denoted as NS-SSCL. The motivation of this unsupervised approach is to make representations of the same attributes and the same rules are as close as possible across different objects and problems, respectively. Critically, the mapping from the neural embeddings of object attributes and the symbols used for rule logic execution can be further established by virtue of their stable representations, even though these representations learned by SSCL are not necessarily aligned well with ground truths in the tasks due to the unsupervised nature. Notably, the current model relies on the probability codes of the discrete symbols of object attribute values (Figure 1) as similar as used in the probabilistic abduction and execution (PrAE) model (Zhang et al., 2021). However, the PrAE model used supervisions of the correct and incorrect answers from the candidate panels, and also relies on the ground truth of rules contained in each instance as auxiliary annotations. In this work, without these ample supervisions, NS-SSCL obtains state-of-the-art performance accuracy in the unsupervised models on the context of instance, but without the candidates, in solving the RPM-like tasks, even better than most of the previous supervised models.

## 2 RELATED WORK

**Object representations** It is critical to correctly recognize the exact object attributes in visual reasoning tasks, as the latent rules governing the context are defined by these features. Although deep neural networks are versatile to fit any desired function constrained by the loss function, the neural embeddings of latent object representations are too flexible to fit well with the semantics of

object attributes used to form the latent rules, such as types, sizes, and colors in the RAVEN dataset. Instead, these object attributes are often mixed in the embeddings. Recent studies have proposed using variational autoencoder (VAE) (Mańdziuk & Zychowski, 2019; Klein & Nabi, 2020; Higgins et al., 2017; Burgess et al., 2018; Pekar et al., 2020) and neural-vectors (Hersche et al., 2022) to disentangle the blended representations of visual attributes. SSCL has been used to discover better representations by comparing the augmented data from the same or different images (Chen et al., 2020; He et al., 2019). We here leverage this unsupervised method to shaping the neural embeddings to largely comply with the semantic dimensions of defined object attributes in tasks.

**Visual reasoning**   Most of supervised models designed to solve visual reasoning tasks mainly focus on the visual reasoning process (Hoshen & Werman, 2017; Barrett et al., 2018; Zhang et al., 2019b; Zhuo & Kankanhalli, 2020; Mańdziuk & Zychowski, 2019; Zhuo & Kankanhalli, 2021), as the baselines of DL models fail to solve these high-level cognitive tasks. A common motivation for visual reasoning models is to learn relational representations of latent rules governing the task context by maximizing similarity between analogical relations and minimizing similarity between non-analogical relations (Barrett et al., 2018; Zhang et al., 2019b; Jahrens & Martinetz, 2020; Malki'nski & Ma'ndziuk, 2022; Wu et al., 2021; Kiat et al., 2020; Kim et al., 2020). This is achieved by comparing the relational representations of the correct answer with that of the incorrect answers. Instead, it can also directly utilize the relation structures within the context only and with no supervisions. We here leverage the SSCL unsupervised strategy on the contexts of instances to training the rule reasoning module. Differing from the pairwise relations discriminator (PRD) model (Kiat et al., 2020), we here further use SSCL to learn the neural representations of object attributes and additionally map these representations with the symbols used for rule reasoning.

**Neuro-symbolic models**   It arrives at consensus that the NS architecture might be a proper model to provide interpretability and generalizability in solving abstract reasoning tasks (Yi et al., 2019; Mao et al., 2019; Zhang et al., 2021; Ding et al., 2021b). Unlike the monolithic DL models, the NS models are composed of a perception module and a reasoning module (Zhang et al., 2021; Ding et al., 2021b). Nonetheless, it remains challenges to train the NS models in an end-to-end supervised form. Thereby, auxiliary annotations of the latent rules are required to constrain the rule representations in the PrAE model (Zhang et al., 2021). In striking contrast to PrAE, we here use SSCL to separately train the neural representations of object attributes and rules separately in the perception and reasoning modules of the NS model.

## 3   METHODS

The current work here mainly focuses on the learning process, rather than the NS architecture (Figure 1) (Chen et al., 2020). This proposed model put forward advances in AI solving abstract visual reasoning as follows. First, inspired by human cognitive processes of solving these tasks, we are motivated to train a NS model equipped with the general cognitive abilities of visual object attribute recognitions and rule recognitions, both of which are readily acquired in humans prior to solving RPMs (Marcus & Davis, 2020; Fodor & Pylyshyn, 1988; Spelke, 1990; Gopnik et al., 2004; Li & DiCarlo, 2008; Mansouri et al., 2020). To acquire the two general cognitive abilities, we leverage SSCL on the context set, but not the candidate set or the answer in each task instance (Figure 2). Second, we do not require the NS model to have prior knowledge about either the dimension of object attributes or the underlying rule information, including their formats and dimensions (Zhang et al., 2021). This provides the current method with flexibility in solving other visual reasoning tasks. Third, due to the perception and reasoning modules are separately and sequentially trained, the representation-symbol mapping from the perception module to the reasoning module is easily optimized, even though both are not perfect due to the lack of ground truths. Fourth, the training is not relied on the candidate sets and the selection of answer from candidates is generative from the context. In brief, training the current model on the context of instance consists of four sequential processes (Figure 2): (1) Pretraining for semantic concepts; (2) Rule Induction Network; (3) Symbolic rule knowledge; (4) Fine-tuning, while solving the tasks consists of three processes (Figure 1): (1) Object attribute perception; (2) Rule reasoning; (3) Answer generation (see implementation details in Appendix B).

## 3.1 Pretraining for semantic concepts

For the RAVEN task (Matzen et al., 2010; Zhang et al., 2019a), we use an object-based convoluted neural network (CNN) to parcel the objects into attributes in the perception module (Figure 1). The attributes include existence of objects, types, sizes and colors, while the representations of types, sizes, and colors are subject to whether existence of objects is true. We assume that the model has such prior semantic concepts about type, size and color. we thus generate standard forms of objects for types, sizes, and colors, respectively (Figure 2A). For example, a standard form for the color attribute is a circle of standard size which has the same color as the original object. So, if two objects share the same color, they should have similar standard forms of color (Chen et al., 2020). For each attribute, we train the perception module to have similar representations for objects and their corresponding standard forms. Further, the perception module distinguishes different attribute values. The loss function has a similar form as in contrastive learning except that the output of perception module is coded in a probability form:

$$Loss_c(I) = \frac{1}{N}\sum_{i=1}^{N} -log\Big(\frac{D(p_i, q_i, \tau_a)}{\sum_{j=1}^{N} D(p_i, q_j, \tau_a)}\Big), Loss_s(I) = \frac{1}{N}\sum_{i=1}^{N} -log\Big(\frac{D(q_i, q_i, \tau_a)}{\sum_{j=1}^{N} D(q_i, q_j, \tau_a)}\Big)$$

$$Loss_h(I) = -log\Big[H\Big(\frac{\sum_{i=1}^{N} q_i}{|\sum_{i=1}^{N} q_i|}\Big)\Big], \text{ with } N = |I|, \ D(p, q, \tau) = e^{p \cdot q/\tau}$$

$$Loss_a(I) = \lambda Loss_s(I) + (1-\lambda)Loss_c(I) + \mu Loss_h(I)$$

$$(1)$$

where $I$ is the set of inputs in the batch, $a$ is a certain attribute, $p_i, q_i$ are the probability output of the $i$th original object in $I$ and corresponding standard form respectively, $\tau$ is the temperature parameter, $\lambda, \mu$ are weight parameters and $H$ means entropy. The term $Loss_c$ is the contrastive loss between original objects and standard forms, the term $Loss_s$ is the contrastive loss with standard forms, and the term $Loss_h$ is to avoid a trivial situation that $p_i \cdot p_j = 1$ holds for all inputs.

In addition, the chosen part of image fed to the perception module may contain no object. This would affect the object-based pretraining and later the reasoning. So, we add a term for existence to the loss and only use inputs with existence of objects to calculate the attribute loss as listed above. The existence loss is calculated by negative log-likelihood and the labels are obtained by simple image processing methods. The final loss is:

$$Loss = Loss_{type}(I_e) + Loss_{size}(I_e) + Loss_{color}(I_e) + Loss_{exist}(I)$$
$$\text{with } I_e = \{i \in I | \text{ an object exists in } i\}$$

$$(2)$$

For simplicity, the perception module is no longer updated after pretraining. Different objects' attributes within a panel are further integrated to form the panel information.

For the V-PROM task (Teney et al., 2020), we used a ResNet (He et al., 2015) network pretrained on ImageNet (Russakovsky et al., 2014) and further finetuned it with supervision. However, the corresponding order of supervised labels of object attributes is randomly shuffled from the ground truths. That is, similar to the perception module in the RAVEN task, the perception module also only knows whether two attribute values are same or not, but do not know the exact attribute value (ground truth).

## 3.2 Rule Induction Network

We also build a model having the ability of distinguishing rules through SSCL (Figure 2B). The unique feature of rules governing the RPM-like tasks is that the latent rule for each attribute is row-wised and consistent across the rows in the matrix of a context instance. Thereby, the same attribute should share the same rule across the first two rows of each instance (intra-problem rule analogy as the positive examples), but unlikely share the same rule across the rows paired between different problems (inter-problem rule non-analogy as the negative examples) (Wu et al., 2021; Kiat et al., 2020). We then use this meta-rule knowledge about the task structure to constraint and induce rules. Differing from previous supervised models, we do not require the model to precisely produce representations of the rules, but only require it to discriminate the rule relationship of a given pair

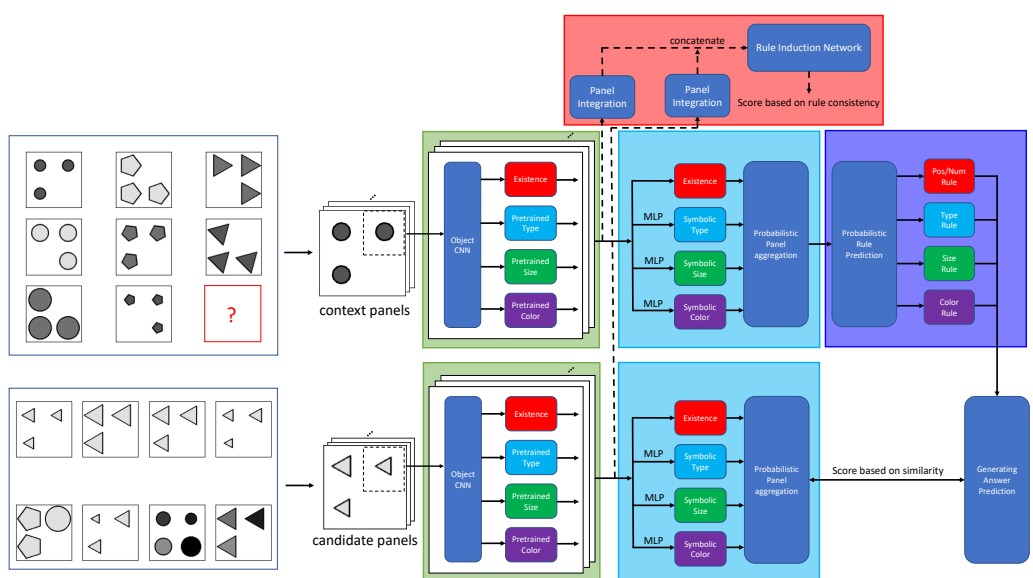

Figure 1: **A schematic of the model architecture.** See the main text for details.

of rows. For training, we only use the first two rows in each context of instance. Therefore, we have $2N$ rows for a batch of size $N$, and have $2N \times 2N$ pairs of rows. We concatenate the perception output of each pair as the input of RIN. If the two coupled rows come from the same instance, then the pair is labeled as 1, indicating they share the same rule, otherwise, it is 0. The RIN is a MLP for each attribute $a$ , and the loss function is weighted L1 as follows:

$$Loss_a^{RIN} = \sum_{i,j=1,2} w_{i,j,\delta_{x,y}} \cdot \left| f_a(\text{row}_a^{x,i}, \text{row}_a^{y,j}) - \delta_{x,y} \right|_1 \tag{3}$$

where $f$ stands for RIN, $i, j$ are row indexes, $x, y$ are RPM indexes, $w_{i,j,\delta_{x,y}}$ are weight parameters and $\delta$ means the Kronecker delta function.

On the basis of the relationships across the rows, we build up an unweighted and undirected graph network for each attribute where a node stands for a context of instance and an edge indicates a shared rule between a pair of instances.[1] Operationally, only a subset of samples (e.g., 512) are chosen to build up the graph. An edge is added into the graph when a pair of the first rows from two different instances have RIN outputs above the threshold 0.5:

$$\mathcal{G}_a = (V, E_a)$$
$$V \subseteq \text{RPMs}, \ E_a = \{(x, y) \in V \times V \mid x \neq y, f_a(\text{row}_a^{x,i}, \text{row}_a^{y,j}) > 0.5 \ \forall i, j = 1, 2\} \tag{4}$$

To classify the rule structures, we use the label propagation method (Raghavan et al., 2007) of community detection to find the communities, or clusters, in the graph within which the nodes are more frequently connected to one another and less to the rest of the network. Notably, the current contrastive learning approach by RIN does not directly result in discriminations of the rules, but the problem identities. The facts that different attribute values with the rule of "distribution three" or "union" do not appear in the same instance, lead to the problems with such a rule are isolated with one another in the graph (Figure 4).

### 3.3 SYMBOLIC MAPPING

The model can detect the rule categories of each attribute. But since it does not know the order or exact value of each attribute, it cannot specify the exact rule, particularly the progression or

---

[1]In fact, we can directly use the RIN to solve RPMs with a method similar to pairwise relations discriminator (PRD) (Kiat et al., 2020) proposed in the previous study to compare the similarity of relations between rows, namely we compare the rule-consistency between the third row and the first two rows by individually adding each candidate into the third row of the RPM context.

arithmetic rules. Generally, a rule of each attribute can be viewed as a set of ordered tuples. For example, the arithmetic plus rule can be viewed as:

$$r_+ = \{(1,1,2),(1,2,3),(1,3,4),\dots\} \tag{5}$$

To identify the exact rules, we further map the series of representations of each attribute in the perception module to a set of tokens or symbols used in the rule reasoning module. To do so, we train a MLP to map the neural representations to the symbols used for reasoning the rules. Specifically, we run all possible rules with the set of attribute values under the assumed symbolic knowledge, we then calculate the probability of rules by aggregating all probabilities of attribute value distributions. For example, the probability that the panels of a row $(a,b,c)$ follows the arithmetic plus rule on a certain attribute can be calculated as:

$$P(r_+|a,b,c) = \sum_{(x,y,z)\in r_+} P(a=x)P(b=y)P(c=z) \tag{6}$$

Hence, what we do is just to properly assign the rules to corresponding problem communities found in the graph by maximizing the probability of the assignments, i.e., the probability that all instances follow corresponding assigned rules (Figure 2C). In theory, an improper assignment may cause conflicts and thus lead to a higher loss value. We then train a MLP to map the representations of object attributes to the corresponding symbols.

## 3.4 FINETUNING

Further, we use rules predicted by symbolic reasoning as pseudo-labels to finetune the weights of the MLP that maps the neural representations to the symbols (Figure 2D). To predict the rule, we normalize the rule probability and choose the rule with highest normalized probability as in PrAE (Zhang et al., 2021). Normalizing rule probability means both to balance the numbers of possible situation among rules and to make the sum of rule probability 1.

$$P_{norm}(r) = \frac{Q(r)}{\sum_{r'\in E} Q(r')} \text{ with } Q(r) = \frac{P(r)}{|r|} \tag{7}$$

where $E$ denotes the set of possible rules. The loss function is the negative log-likelihood of the predicted rules. This approach of self-correction usually further improves performance.

## 3.5 ANSWER GENERATION

Finally, the model also predicts the rules of each attribute through symbolic reasoning, and accordingly generates a predicted probability distribution of attributes. Meanwhile, the probability distributions of object attributes of the candidates are also obtained by the model. We then compare the Jensen-Shannon divergence (JSD) between the predicted distributions with those of the candidates. The candidate with smallest divergence is then selected as the answer.

## 4 EXPERIMENTS

### 4.1 EXPERIMENTAL SETUP

We test NS-SSCL on the RAVEN with the three benchmarks of RAVEN, I-RAVEN and RAVEN-FAIR (Zhang et al., 2019a; Hu et al., 2021; Benny et al., 2020). The three benchmarks share the same contexts, but different candidate sets. I-RAVEN and RAVEN-FAIR are later designed to avoid the drawbacks in the original RAVEN benchmark in that the candidate panels have shortcuts to identify the correct answers even without context information (Hu et al., 2021). Because our training processes do not access the candidate sets, the training processes had no differences in these three benchmarks. Also, we test NS-SSCL on a portion of V-PROM dataset with object counts as the attribute for rule reasoning (Appendix F).

### 4.2 EXPERIMENTAL RESULTS

**Evaluations of general performance in I-RAVEN**   First, we evaluate the proposed model's performance on I-RAVEN in comparison with some representative supervised and unsupervised models. To the best of our knowledge, the NVSA (Hersche et al., 2022) and SCL (Wu et al., 2021)

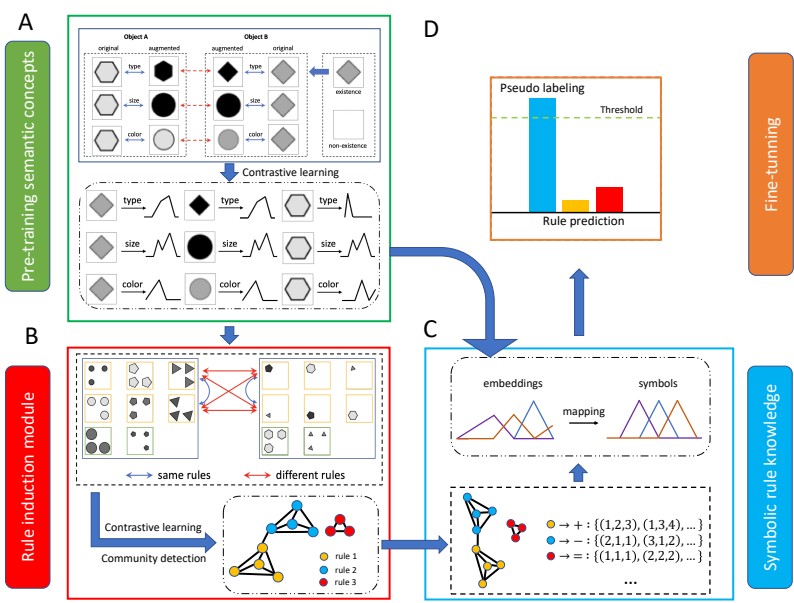

Figure 2: **The components of NS-SSCL.** (A) Pretraining semantic concepts of objects; (B) Building-up Rule Induction Network; (C) Constructing symbolic rule knowledge; (D) Fine-tuning.

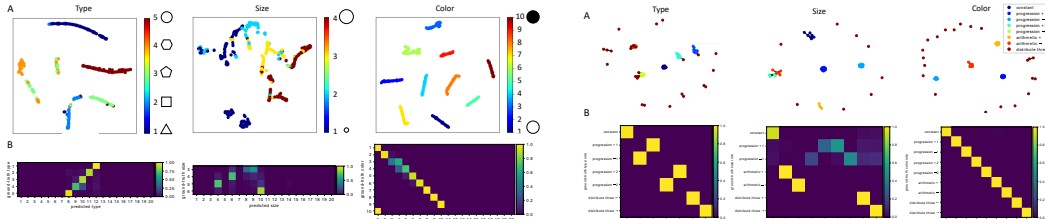

Figure 3: **The representations of object attributes in the perception module.** (A) The embeddings of object attributes (t-SNE) ; (B) The confusion matrix of attributes.

Figure 4: **The representations of rules in the rule reasoning module.** (A) The graph built by RIN and the communities with predicted rules ; (B) The confusion matrix of rules.

models are the currently two state-of-the-art supervised models (Malki'nski & Ma'ndziuk, 2022). We also compare with the PrAE model (Zhang et al., 2021), as our model shares similar probability codes.[2] The unsupervised models used in RAVEN are few, we here compare with the PRD model (Kiat et al., 2020). Table 1 shows the performance of different models. On average, our unsupervised model achieves 95.5% accuracy on I-RAVEN, 39.6% higher than the PRD model. Even in comparison with the supervised models, our unsupervised model achieves performance better than SCL and PrAE, particularly in 2×2 Grid, 3×3 Grid and O-IG configurations, but not NVSA.

**Evaluations of performance in each sub-process**  We then evaluate the importance of each component in the proposed model. As the four components are sequentially conducted, we can stepwisely add the corresponding component into our model and then assess its importance (Appendix C). Table 2 shows the performance improvement on the three RAVEN benchmarks by sequentially adding each component into the model (Table 5 for details). On average, the performance on the

---

[2]The PrAE models here are trained separately on each configuration. But since PrAE model is hard to train on 3×3 Grid due to limited computational resources, the 3×3 Grid result shown in this paper is the test accuracy of the model trained on 2×2 Grid as in the original PrAE paper (Zhang et al., 2021), which is starred in Table 1.

three benchmarks is similar, but consistently lower in RAVEN than I-RAVEN and RAVEN-FAIR. This is likely due to the candidate panels in RAVEN share more object attributes (Zhang et al., 2019b; Hu et al., 2021), causing the generative answer a little more difficult to distinguish the true one from the other candidates, as similar as in PrAE (Zhang et al., 2021). In details, testing the perception module only achieves average test accuracy close to 50% in I-RAVEN and RAVEN-FAIR. Second, adding the RIN component into the model improves the accuracies about 20%. Please keep in mind that RIN only has concepts about the rule categories, but not the exact rules. Third, after mapping the representations of object attributes to symbols with a MLP, the accuracies further increase over 20%. The two steps of symbolic rule reasoning module increase the accuracies by 40%, indicating their importance in the current model. Finally, finetuning further improves the accuracies.

Table 1: Accuracy(%) of different models on I-RAVEN

| Supervised Method | Avg | Center | 2x2Grid | 3x3Grid | L-R | U-D | O-IC | O-IG |
|---|---|---|---|---|---|---|---|---|
| PrAE | 87.8 | 100.0 | 87.5 | 55.5* | 97.6 | 98.1 | 98.4 | 78.0 |
| NVSA | 98.8 | 100.0 | 99.6 | 96.7 | 100.0 | 100.0 | 100.0 | 95.0 |
| SCL | 95.0 | 99.0 | 96.2 | 89.5 | 97.9 | 97.1 | 97.6 | 87.7 |
| Unsupervised Method | Avg | Center | 2x2Grid | 3x3Grid | L-R | U-D | O-IC | O-IG |
| PRD | 55.9 | 73.1 | 39.9 | 35.3 | 67.3 | 67.3 | 68.1 | 40.6 |
| NS-SSCL(Ours) | 95.5 | 99.1 | 95.7 | 94.5 | 96.4 | 97.5 | 92.4 | 93.1 |

Table 2: Average accuracy(%) of 4 levels of models

| Dataset | PM | PM+RIN | PM+RIN+RK | PM+RIN+RK+FT |
|---|---|---|---|---|
| RAVEN | 35.0 | 57.4(+22.4) | 83.8(+26.4) | 92.3(+8.4) |
| I-RAVEN | 48.3 | 68.5(+20.2) | 89.7(+21.1) | 95.5(+5.8) |
| RAVEN-FAIR | 53.8 | 71.3(+17.5) | 91.5(+20.2) | 96.2(+4.7) |

Table 3: Accuracy(%) of models trained on 2x2Grid only

| Method | Avg | Center | 2x2Grid | 3x3Grid | L-R | U-D | O-IC | O-IG |
|---|---|---|---|---|---|---|---|---|
| NS-SSCL(Ours) | 68.8 | 60.3 | 95.7 | 66.0 | 79.7 | 79.5 | 57.0 | 43.5 |
| PrAE | 77.0 | 90.5 | 85.4 | 45.6 | 96.3 | 97.4 | 63.5 | 60.7 |

Notably, differing from the supervised learning with the auxiliary annotations of object attributes or rules, the neural representations of object attributes (Figure 3) and the rule assignments (Figure 4) are not perfect by the current SSCL method without any supervision. Nonetheless, the model can perform quite well the RAVEN visual reasoning task. This is largely credited by the symbolic mapping.

**Evaluations of generalizability** We further evaluate the generalizability of our proposed model in solving I-RAVEN. First, we examine the performance dependence on the training sample size (Appendix D). Figure 5 illustrates that our model is immune to the reduction of training sample sizes. Even when the number of training samples is reduced to 300, 5% of the full sample size, the average accuracy remains as high as 80.5%. Although PrAE has a similar robustness of performance, the performance of SCL (Wu et al., 2021) is considerably sensitive to the training sample sizes, and reduced to the chance level when the training samples are 10% and 5% of the full samples. Second, we evaluate cross-configuration generalizability for our model (Figure 6). The models trained in the simple configurations of center, L-R, U-D and even $2\times2$ Grid fairly transfer to solve the problems in the other simple configurations, but the models trained in the $3\times3$ Grid, O-IC, and O-IG cannot transfer to solve the problems in simple configurations (Appendix E). Our model in $2\times2$ Grid has roughly similar generalizability as in PrAE (Table 3).

**Tests on the V-PROM task** We further test NS-SSCL on the V-PROM task (Appendix F). We selected to test on the feature of object count, as only this part is valid for all the rules in V-PROM. It

Table 4: Accuracy(%) of models on object count split of V-PROM

| Method | Accuracy |
|---|---|
| Baseline (RN, ResNet + aux. loss) | 55.4 |
| NS-SSCL | 85.6 |

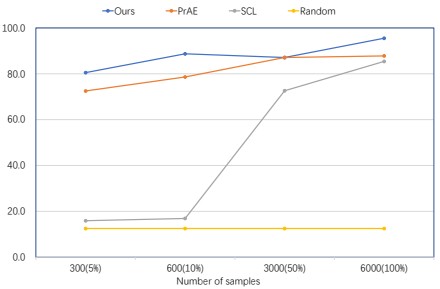

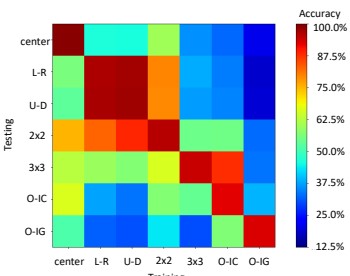

Figure 5: **The test accuracy change with number of samples.**

Figure 6: **Cross-configuraton generalizability.**

turns out that NS-SSCL achieves much better performance (Table 4) than the baseline method using a relational network (RN) (Teney et al., 2020).

## 5 CONCLUSION

In this paper, we present NS-SSCL to obtain state-of-the-art performance in unsupervised models on the RPM-like tasks involving abstract visual reasoning. The previous state-of-the-art results were obtained by learning the task-specific knowledge with supervision. By contrast, NS-SSCL provides an approach to establish the general cognitive abilities in solving the visual reasoning task from the contexts of tasks, but not the task-specific knowledge (Marcus & Davis, 2020; Fodor & Pylyshyn, 1988; Chollet, 2019). Thereby, its performance has strong robustness even for small sample size for training and broad generalizability for cross-configuration tests. The simplicity of this approach, we believe, should afford its broad applications in solving other spatiotemporal reasoning tasks (Chollet, 2019; Spratley et al., 2020). Further, although the SSCL method has been extensively used in many domains, these models share a common drawback of lacking interpretability. We here use SSCL to train a NS model, to provide interpretability and generalizability in solving high-level cognitive tasks, such as RPM-like tasks, in a human-like manner. In our opinion, this is a promising approach towards AGI (Chollet, 2019).

The current model of NS-SSCL also has some important limitations deserved to be improved. First, both the perception and reasoning modules can be pretrained by independent situations and tasks to construct its general cognitive abilities in object and rule recognitions. It remains to explore this potential by training independent tasks and testing on other independent tasks, similar to the models used in NLP (Dong et al., 2018; Dosovitskiy et al., 2020). Second, the NS-SSCL performance might be further improved by jointly optimizing the interactions between the components. Third, more general cognitive abilities and higher-level cognitive function (e.g., metacognition) can be further incorporated into the model to provide more versatile intelligent abilities in solving complex tasks. Finally, the potential applications of the current model should be extensively tested in other abstract visual reasoning tasks.

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

## APPENDIX

## A  DETAILS OF RAVEN DATASET

There are 7 configurations in RAVEN dataset as shown in Figure 7 (Zhang et al., 2019a). The configurations are determined by the different organizations of a variety of objects in each panel (i.e., the cell of the 3×3 matrix in the context). For instance, in the configuration of 2×2 Grid, each panel consists of 2×2 cells and in each cell an object may be occupied or not. The number and position attributes are then determined by the compositions of cells in each panel. On the other hand, in the configuration of Left-Right, each panel consists of the left and right parts, but the two parts are independent, thereby, the rules governing each part of the context panels are also independent.

The rules that govern the RAVEN dataset are normatively classified into (1) **Constant**; (2) **Progression**; (3) **Arithmetic**; and (4) **Distribute Three**. Further, internal parameters are used to diversify the above abstract rules into concrete rules. For instance, **Arithmetic** contains plus or minus, while **Progression** contains increments or decrements of 1 or 2, and **Distribute Three** contains left-revolving or right-revolving. In total, there are 9 possible concrete rules. However, the numbers of possible concrete rules are different for different object attributes.

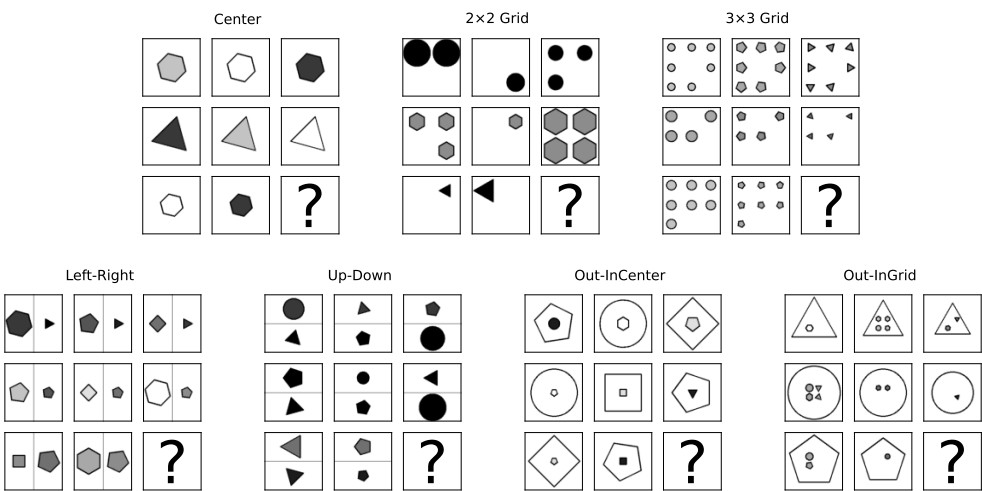

Figure 7: Representative RAVEN problems of all configurations

## B   IMPLEMENTATION DETAILS

**General implementation details on RAVEN**   We separately train the models for the 7 different RAVEN configurations [Center, Left-Right (L-R), Upper-Down (U-D), 2×2 Grid, 3×3 Grid, Out-In Center (O-IC), Out-In Grid (O-IG)]. We train our model on the 6,000 samples in the training dataset and test the model on the 2,000 samples in the testing dataset for each configuration. The models are implemented in PyTorch and run with Intel(R) Xeon(R) Platinum 8272CL CPUs and NVIDIA Geforce RTX 3090 Founders Edition GPUs. The inputs of the object images (originally $160 \times 160$ for a panel) are resized to $32 \times 32$. We have referred to PrAE (Zhang et al., 2021) in parts of our implementation. For instance, in the answer generating part, we follow the idea of executor in PrAE.

**Implementation details in visual perception module**

$$Loss_c(I) = \frac{1}{N} \sum_{i=1}^{N} -log\Big(\frac{D(p_i, q_i, \tau_a)}{\sum_{j=1}^{N} D(p_i, q_j, \tau_a)}\Big), Loss_s(I) = \frac{1}{N} \sum_{i=1}^{N} -log\Big(\frac{D(q_i, q_i, \tau_a)}{\sum_{j=1}^{N} D(q_i, q_j, \tau_a)}\Big)$$

$$Loss_h(I) = -log\Big[H\Big(\frac{\sum_{i=1}^{N} q_i}{|\sum_{i=1}^{N} q_i|}\Big)\Big], \text{ with } N = |I|, \; D(p, q, \tau) = e^{p \cdot q / \tau}$$

$$Loss_a(I) = \lambda Loss_s(I) + (1 - \lambda)Loss_c(I) + \mu Loss_h(I) \tag{8}$$

For visual perception pretraining, the batch size of the RAVEN problems $N$ is 32 and only the 8 context panels are used. The number of object images used for pretraining in each batch depends on the configurations. For instance, in configuration of 2×2 Grid, there are $32 \times 8 \times 4 = 1024$ objects in each batch. The weight parameter $\lambda$ is 0.6, $\mu$ is 0.01. The temperature parameter $\tau$ is 0.1 for the attribute of type, and 0.05 for the attributes of size and color. The object-based visual perception modules are trained for 100 epochs. The dimensions of probabilistic representations of object attributes are 20, while the actual categories or values of the type, size and color attribute are 5, 4, and 10, respectively.

**Implementation details in rule reasoning module**

$$Loss_a^{RIN} = \sum_{i,j=1,2} w_{i,j,\delta_{x,y}} \cdot \big|f_a(\text{row}_a^{x,i}, \text{row}_a^{y,j}) - \delta_{x,y}\big|_1 \tag{9}$$

For training rule induction in RIN, the batch size $N$ is 8. Inputs with the same row index, e.g., both the first row, are weighted by 0.1. Besides, positive samples, i.e., rows from the same RAVEN

problem, are weighted by $N - 1$. The network is trained for 100 epochs. In search for best rule assignment, the learning rate of ADAM optimizer is set to be 0.1, and the model is optimized for 20 steps before evaluating each assignment. Notably, longer training or lower learning rate should lead to more precise evaluation but also more compute time. After the best assignment is determined, we reduce the learning rate to be 0.01 and conduct training for 1,000 steps. The number of vertices in the graph network, i.e., the batch size, is 512 in common setting but is limited to $\min\{512, 0.8n\}$ in few-shot setting with $n$ samples in total.

**Implementation details in finetuning**    As for finetuning, the learning rate is 0.01 and gradients are clipped into the range [-1,1]. The batch size is 256 and the number of epoch is 10. For each batch, we first predict the rules and then optimize the MLP mapping representation to symbols accordingly for 30 steps. The rule prediction with normalized probability less than 0.9 or absolute probability less than $10^{-10}$ are weighted with 0.

**Implementation details in panel information integration**    The panel information integration process is slightly different for different procedures. The probability of the type, size, or color attribute is existence-weightedly averaged for rule induction as follows:

$$p_a^{panel} = \frac{\sum_{obj} p_a^{obj} p_{exist}^{obj}}{\sum p_{exist}^{obj}} \tag{10}$$

Instead, the log probability is existence-weightedly averaged for optimizing rule probability:

$$\log(p_a^{panel}) = \frac{\sum_{obj} \log(p_a^{obj}) p_{exist}^{obj}}{\sum p_{exist}^{obj}} \tag{11}$$

While testing, the log probability of objects is averaged in a probabilistic-marginalizing-style, as similar as PrAE (Zhang et al., 2021):

$$p_a^{panel} = \sum_{exist} p_{exist} \cdot \exp\left(\frac{\sum_{obj} \log(p_a^{obj}) exist_{obj}}{\sum exist_{obj}}\right) \tag{12}$$

where $exist$ is a binary vector describing the existence of objects.

As for the integration of existence, RIN simply concatenates the object-based prediction in each panel. While probabilistic integration is used to obtain the position and number information of each panel during testing. Notably, for independent components, for instance, in the U-D configuration, the object information is not integrated and is processed independently.

## C    PERFORMANCE OF THE FOUR LEVELS OF MODELS ON EACH CONFIGURATION

Table 5 details the test accuracy on each configuration for our 4 levels of models described in the main text. It is much appreciated that the accuracy is greatly improved by the rule reasoning procedures of RIN and rule assignments in each configuration, respectively.

## D    VARYING THE BATCH SIZE WHILE PRETRAINING THE PERCEPTION MODULE

In SSCL (Chen et al., 2020), the performance in image categorization is significantly dependent on the batch size. The larger the batch size, the higher the performance. To test whether our proposed model of NS-SSCL is also dependent on the batch size used for visual perception module, we train the module with different number of batch sizes. Notably, the batch size is the number of the RAVEN problems, rather than the contained objects. For sake of simplicity, we here merely test on the configuration of Center on I-RAVEN. On this configuration, there are 8 objects in each RAVEN

Table 5: Accuracy(%) of 4 levels of models for each configuration

| Dataset | Config | PM | PM+RIN | PM+RIN+RK | PM+RIN+RK+FT |
|---|---|---|---|---|---|
| RAVEN | Avg | 35.0 | 57.4(+22.4) | 83.8(+26.4) | 92.3(+8.4) |
| | Center | 31.8 | 60.1 (+28.4) | 91.4 (+31.3) | 98.6 (+7.2) |
| | 2x2Grid | 60.4 | 73.3 (+13.0) | 89.5 (+16.2) | 92.9 (+3.4) |
| | 3x3Grid | 48.3 | 55.5 (+7.2) | 90.3 (+34.8) | 91.7 (+1.5) |
| | L-R | 18.0 | 51.8 (+33.9) | 76.3 (+24.5) | 94.0 (+17.8) |
| | U-D | 22.1 | 67.1 (+45.0) | 78.7 (+11.6) | 93.9 (+15.2) |
| | O-IC | 27.1 | 36.7 (+9.6) | 76.5 (+39.9) | 86.0 (+9.5) |
| | O-IG | 37.4 | 57.5 (+20.1) | 84.3 (+26.8) | 88.9 (+4.6) |
| I-RAVEN | Avg | 48.3 | 68.5(+20.2) | 89.7(+21.1) | 95.5(+5.8) |
| | Center | 43.9 | 69.9 (+26.0) | 95.6 (+25.7) | 99.1 (+3.5) |
| | 2x2Grid | 63.7 | 80.1 (+16.4) | 93.0 (+12.9) | 95.7 (+2.8) |
| | 3x3Grid | 52.0 | 63.3 (+11.3) | 92.8 (+29.5) | 94.5 (+1.8) |
| | L-R | 38.9 | 60.7 (+21.8) | 82.7 (+22.0) | 96.4 (+13.8) |
| | U-D | 41.9 | 82.1 (+40.2) | 88.4 (+6.3) | 97.5 (+9.2) |
| | O-IC | 46.9 | 52.2 (+5.4) | 85.8 (+33.6) | 92.4 (+6.6) |
| | O-IG | 51.2 | 71.7 (+20.5) | 89.8 (+18.1) | 93.1 (+3.4) |
| RAVEN-FAIR | Avg | 53.8 | 71.3(+17.5) | 91.5(+20.2) | 96.2(+4.7) |
| | Center | 47.1 | 69.6 (+22.5) | 95.6 (+26.0) | 99.3 (+3.8) |
| | 2x2Grid | 75.8 | 84.7 (+8.9) | 94.5 (+9.8) | 95.4 (+1.0) |
| | 3x3Grid | 66.2 | 72.2 (+6.0) | 95.7 (+23.6) | 96.2 (+0.5) |
| | L-R | 39.5 | 64.1 (+24.6) | 86.5 (+22.4) | 96.9 (+10.5) |
| | U-D | 43.0 | 82.4 (+39.4) | 89.3 (+6.9) | 97.1 (+7.8) |
| | O-IC | 45.5 | 53.3 (+7.8) | 86.5 (+33.2) | 93.8 (+7.3) |
| | O-IG | 59.9 | 73.1 (+13.3) | 92.8 (+19.7) | 95.0 (+2.2) |

problem used for training the visual perception module. As a matter of facts, the performance is insensitive to the batch sizes used for training (Table 6). This is different from SSCL used in visual categorizing tasks. As we have shown in Table 5, the high accuracy of the model is much relied on the backend rule reasoning module.

Table 6: Accuracy(%) of models with different pretraining batch sizes on Center

| Batch size | 1 | 2 | 4 | 8 | 16 | 32 | 64 |
|---|---|---|---|---|---|---|---|
| Accuracy | 95.1 | 96.4 | 95.6 | 97.4 | 98.9 | 99.1 | 98.1 |

# E  DETAILS OF CROSS-CONFIGURATION TESTS

In the configurations like Center, there is no chance that no object exists. Thus, the models trained on the Center configuration cannot learn such an idea of existence. This would lead to extremely poor performance when the models are used to test on the configurations like 2×2 Grid. For this reason, the cross-configuration tests in this study actually rely on the existence prediction from the model that is trained on the corresponding configuration. Notably, a significant factor that affects cross-configuration testing performance is the dissimilarity between objects in different configurations, partially because of different resolutions. Figure 8 illustrates representative examples of the differences between objects in the configurations of Center and 2×2 Grid. The difference is most significant at the boundaries of objects. Hence, it seems that the models learn the configuration-dependent semantic concepts about the objects.

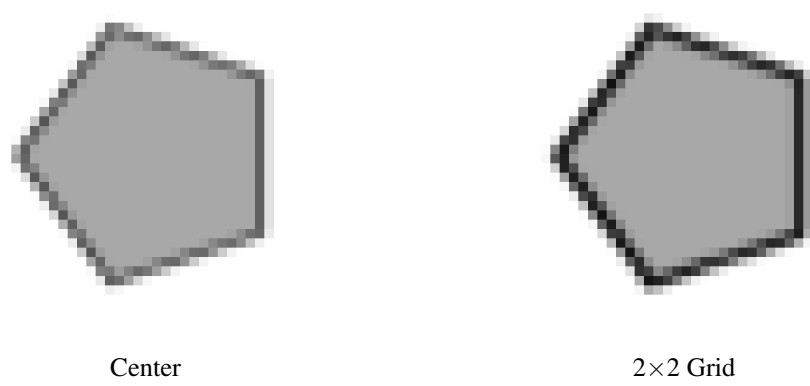

Center                                          2×2 Grid

Figure 8: Example objects in the Center and 2×2 Grid configurations

## F  DETAILS OF V-PROM DATASET

V-PROM is an RPM-like task in which the visual objects are natural images (Figure 9). In this task, there are four types of rules (**And**, **Or**, **Union**, **Progression**). These rules are applied on three dimensions of **Object Attribute**, **Object Category**, and **Object Count**. Only in the dimension of **Object Count** all of the four rules can be applied. For simplicity, we then test on the **Object Count** split of the V-PROM benchmark. We further split the data into the training set and testing set as a ratio of 7 : 3.

Since natural images are used in the V-PROM benchmark, the SSCL approach used in the RAVEN benchmark cannot be directly applied. Instead, we use supervised but label-shuffled samples to finetune a pretrained ResNet model (see 3.1). We also use the community detection method to recognize the clusters of different rules (Figure 10). In following part of incorporating symbolic rule knowledge, slight additional supervision of cluster rules is applied to speed up training.

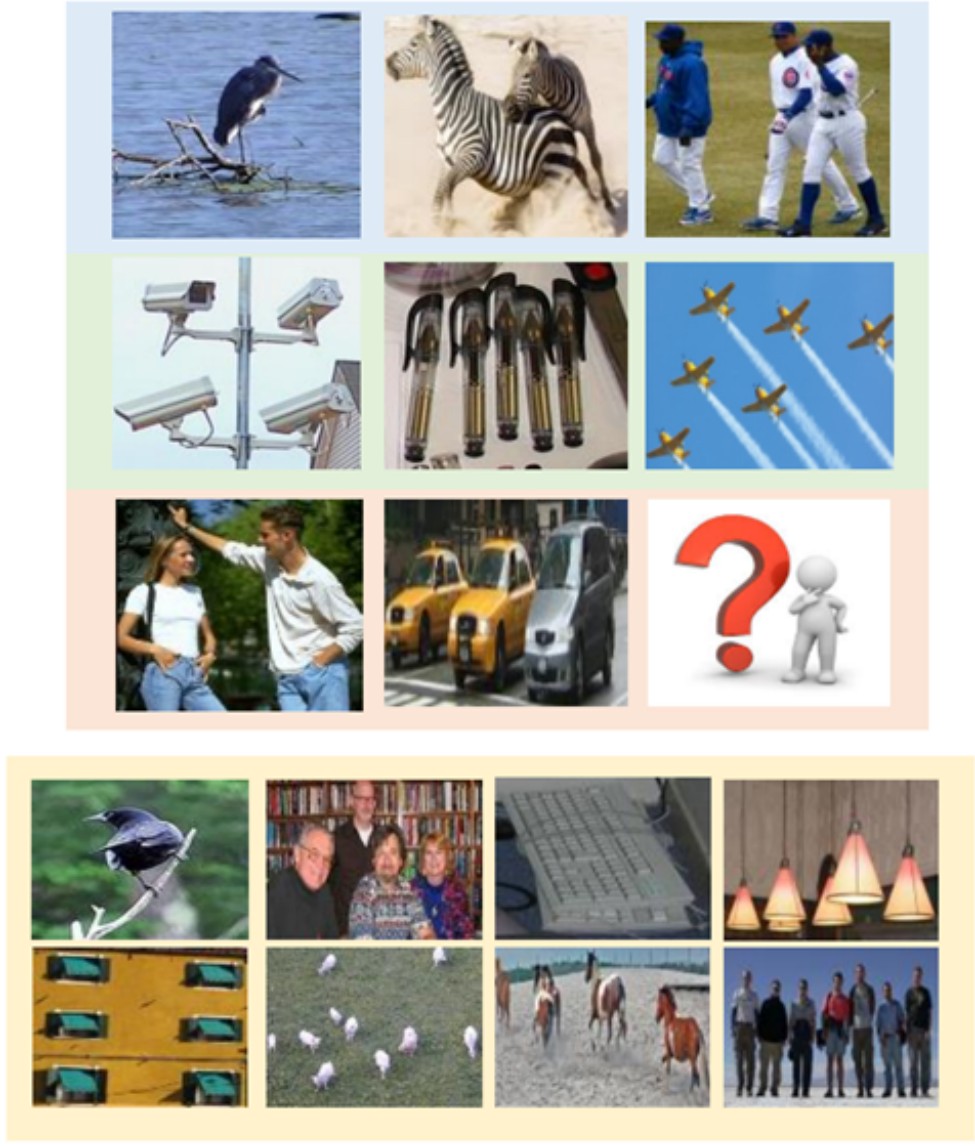

Figure 9: An example of the object count split in the V-PROM benchmark from the original paper (Teney et al., 2020). The correct answer is the second candidate.

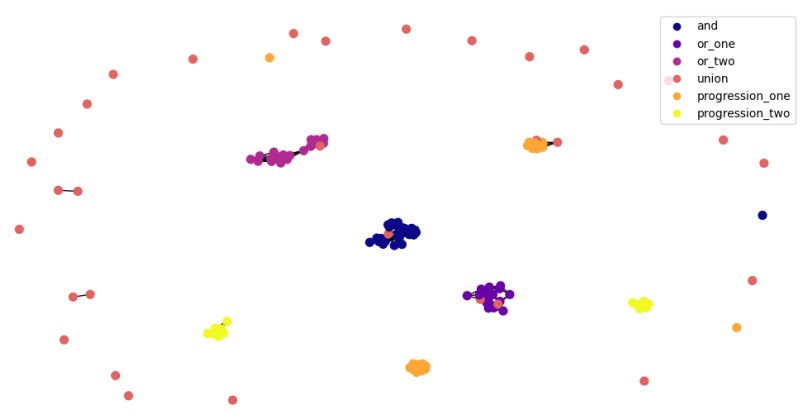

Figure 10: The graph built by RIN and the communities with predicted rules in the V-PROM bench-mark.

