# OpenReview forum: "Abstract Visual Reasoning by Self-supervised Contrastive Learning"
_ICLR.cc/2023/Conference — Submitted to ICLR 2023_

### Official Review · Reviewer_QQJu · 2022-10-16

**Confidence:** 3
**Correctness:** 2
**Technical Novelty And Significance:** 2
**Empirical Novelty And Significance:** 2
**Recommendation:** 3

**Clarity, Quality, Novelty And Reproducibility:**

**Clarity**

The paper provides a very high-level description of the whole framework, leaving out important details, including the underlying assumptions on supervision, the interaction between the different stages during training and inference and precise formal details on the definitions of the losses. Therefore, I suggest the authors to include those assumptions, to provide an algorithmic table summarising the different training stages and the ones used during inference and to give more precise definitions of the losses used to train the model.

**Quality**

As already mentioned in the part about weaknesses, the paper lacks important related work. Additionally, the main claims of the paper are not well supported by the experiments. Also, the obtained results seem to be task/data dependent. Indeed, it remains unclear what is the extent of validity of the proposed approach to tasks other than the RAVEN-based benchmarks.

**Originality/novelty**

The idea of introducing self-supervised learning for abstract visual reasoning is to my knowledge novel and original.

**Strength And Weaknesses:**

Please find a list of strengths and weaknesses and subsequently more detailed comments with actionable points.

**STRENGTHS**
- Solving RPMs with reduced supervision is a relevant and timely challenge.
- Introducing the use of contrastive self-supervised learning is novel in the context of RPMs.

**WEAKNESSES**
- Imprecise/not well supported claims. The claim that the proposed model is unsupervised is imprecise. The authors should clearly state the underlying assumptions on the available supervision and its forms used in the different stages of training. Additionally, the claim that the model is a novel neuro-symbolic framework is imprecise. The authors should clearly specify in which aspect the model can be regarded as a neuro-symbolic approach.
- The proposed solution is tailored to the specific RAVEN problems and it’s not clear how well the strategy can generalise to other settings. Therefore the scope is limited.
- Important related work is missing and should be discussed.
- Experimental comparisons with unsupervised baselines is not fair and the comparisons with supervised ones do not support the superiority of the approach

**COMMENTS and ACTIONABLE POINTS**

The main claims of the paper are vague and not precise. Regarding the unsupervised aspect, it is unclear to me at which level the authors consider their method as an unsupervised strategy. To be more specific:
- When learning attribute level representations, supervision is introduced in at least two ways, namely when augmenting data based on attributes (this relies on the assumption that the attributes are known and the user have full control on the attributes during augmentation) and when relating to attributes of objects/images in the same task (indeed this relies on the assumption that images contain objects with similar properties). Clearly, these aspects are an important limitation of the proposed approach and they should be mentioned and discussed.
- When learning task-level representations, it is unclear which supervised information is used to learn the mapping from attributes to task-specific symbols as the rules seems to be given during training.
Consequently, I suggest to explicitly mention about the underlying assumptions on supervision and clearly state in which stage/aspect the method is considered unsupervised. One can consider (i) to use a table to highlight the similarities and differences in terms of supervision compared to the competitors (e.g. PRD, PrAE) and (ii) to be more precise when defining the equations (1,2,3,6) as lacking such information.
Regarding the neuro-symbolic aspect. In which sense, can the proposed solution be considered a neuro-symbolic framework? The work circumvents the problem of integration between a neural and a symbolic module by proposing a multi-stage procedure, without even mentioning it.

The work seems to be specifically tailored to the experimental setting of RAVEN, which is in stark contrast with the main claims of the paper .  More concretely, it seems that the proposed solution introduces assumptions specific to RAVEN, including the division of image into cells containing a single object, the fact that objects in an image share similar attributes. Furthermore, it seems that several hyper parameters need to be properly tuned, including all the weights for the losses in Eq. 1 and Eq. 2, together with the other hyper parameters required from the different stages. How are these hyper parameters influencing the performance of the proposed approach?
Therefore, how well can the proposed strategy generalise to other abstract visual reasoning tasks, like for instance the ARC benchmark (from Francois Chollet)? How many changes are required in order to ensure a proper level of generalisation in such case?

Important related work is missing. The authors should discuss the relation with respect to neuro-symbolic approaches and object-centric representation learning. Please, refer to [1]-[2] for two good starting points.

As a result of the above-mentioned issues, the experimental comparison with the unsupervised baseline on RPMs is not fair. Additionally, in which sense is the proposed strategy superior to supervised approaches (as mentioned in the abstract and in other parts of the paper)? For instance, by looking at Table 1 in the experimental section, I’m not able to see a convincing improvement over the competitors.

[1] Neural-symbolic learning and reasoning: A survey and interpretation. 2022. Neuro-Symbolic Artificial Intelligence: The State of the Art

[2] Neural Production Systems. 2021 NeurIPS


**Summary Of The Paper:**

The paper focuses on providing a learning strategy tailored to tackle the problem of Raven’s Progressive Matrices (RPMs). The main idea consists of (i) distinguishing between attribute-level (determining the shape, color, size and presence of an object) and task-level (associating a symbol to each combination of object attributes from a row sequence of images) representations and (ii) devising a multi-stage strategy to learn them. Specifically, the attribute-level representation is learnt in a contrastive fashion by leveraging two pieces of information, namely using object pairs generated through attribute intervention and by comparing attributes between image pairs from same and different RPM tasks. The task level representation is learnt in a supervised fashion to distinguish between the different RPM rules. Experiments on RAVEN benchmarks show the comparable performance of the proposed strategy with state-of-the-art supervised approaches.

**Summary Of The Review:**

The paper proposes a multi-stage strategy to tackle abstract visual reasoning tasks based on RPMs. While the idea of using self-supervised contrastive learning is original in this context, the work is preliminary as it requires additional major effort to overcome the highlighted weaknesses.

---

### Official Review · Reviewer_mdUd · 2022-10-23

**Confidence:** 5
**Clarity, Quality, Novelty And Reproducibility:** The paper is not clearly written, of …
**Correctness:** 2
**Technical Novelty And Significance:** 2
**Empirical Novelty And Significance:** 2
**Recommendation:** 3

**Strength And Weaknesses:**

The paper performs thorough evaluation on the chosen problems and is well-motivated. The experimental part details a wide spectrum of the learning landscape and shows different aspects of the model itself compared to others.

My major concern on the work is novelty. PRD is an established work that shows very good performance even when unsupervised. It also uses contrastive learning on the rule induction network and can directly reach answers. From this perspective, using unsupervised learning for facilitation is not new any longer. Besides, the work largely follows PrAE, but only turns the the original supervised method into a self-supervised one with bells and whistles. It's interesting to see it but considering the two aspects mentioned above, the contribution is only incremental.

The paper seems to intentionally confuse reviewers. For one thing, PrAE is a supervised method, so using the intermediate results of PrAE should not be considered unsupervised. The authors continually claim state-of-the-art, while on the experiments, the method fares worse than existing ones.

The manuscript is poorly presented. Can you use more illustrative figures to explain the idea and properly typeset your equations? The equations are so confusing and needs significant rewording to help explain.

**Summary Of The Paper:**

A new model called NS-SSCL, neuro-symbolic self-supervised contrastive learning, is proposed for solving abstract reasoning problems in RAVEN and V-PROM, two abstract reasoning problems used for evaluating machine intelligence. The NS-SSCL method leverages SSCL to learn visual representation for attribute disentanglement, a rule induction network to build community of rule examples self-supervisedly, and finally pseudo labels from the PrAE engine to learn a mapping from representation to symbols. The majority of the model is based on the PrAE workflow and turned into a self-supervised method. In experiments, the authors show that NS-SSCL achieves improved performance compared to certain supervised baselines.

**Summary Of The Review:**

I'm very concerned about the novelty of the work and the presentation of the manuscript. See above for details.

---

### Official Review · Reviewer_Naed · 2022-10-24

**Confidence:** 4
**Correctness:** 2
**Technical Novelty And Significance:** 2
**Empirical Novelty And Significance:** 2
**Recommendation:** 3

**Clarity, Quality, Novelty And Reproducibility:**

The paper presentation is Okay. As I mentioned, it will be good if the authors can discuss in detail the exact amount of supervision they have. Also, there have been several arguments about "human cognition" in the abstract/intro, which should be supported with references.

**Strength And Weaknesses:**

The paper combines neuro-symbolic reasoning methods and self-supervised contrastive learning (SSCL), which indicates an important future direction for neuro-symbolic methods. Specifically, a typical issue with neuro-symbolic methods is that they require additional supervision or strong domain-specific knowledge. Leveraging patterns in existing data to automatically discover such prior knowledge and reduce the amount of annotations needed, is an interesting idea. I acknowledge the contribution of this paper, given that the authors have successfully demonstrated this idea in the RPM domain. Although the domain is toy-ish, I still think these results are promising.

However, I think the current manuscript is still not ready to be published at ICLR due to the following limitations of the methods.

The most critical concern I have about this paper is the exact "supervision" and/or "assumptions" used by the proposed framework. The authors have emphasized a lot in the abstract and intro that a critical issue of existing neuro-symbolic reasoning methods is that "they rely on supervised learning and auxiliary annotations," but the paper has failed to illustrate the advantage of the method in this regard. Specifically,

1. the model indeed requires annotations for the visual recognition modules (size, shape, color) in the first step, because the contrastive loss is computed with respect to "prototypical objects."
2. actually, compared to most existing frameworks for solving RPM tasks, the paper requires a given object detection model. (This point is minor to my score but I think the authors should be explicit about that)
3. the model requires very specific domain-specific knowledge when they are learning "symbol" grounding. Specifically, they require a set of symbolic set that explains all possible structures of models (step 3).

Second, I would not consider the SSCL part in step 2 really "self-supervised" and "domain-independent." Specifically, this training objective is basically the data generation rule of the RPM dataset. That is, the author is exactly training the model using the task objective on RPM (of course, using only the first two rows). It is very unclear how SSCL can be generalized to other visual reasoning tasks.

Third, steps 3 and step 4 seem a bit unmotivated. For example, why do we want to extract symbolic labels from data? Is it for interoperability? Is it for better performance?

Step 3 seems to be a very hard multiple-instance learning problem. How hard is it in this domain?

Also this paper misses some strongly relevant works, especially the work on "perception and grammar" learning:

- Closed Loop Neural-Symbolic Learning via Integrating Neural Perception, Grammar Parsing, and Symbolic Reasoning. Li et al. In ICML 2020.

Meanwhile, this paper seems to be conceptually related to the SCL work (Wu et al). For example, they also study how to use neural networks to learn meta-level rules from data. I would like to see a detailed comparison between this work and SCL.

Minor questions:

1. In step 2, how would the model handle ambiguities in progression rules? For example, the observation "123" can be interpreted as "progression" or "arith addition."
2. I think the results in Table 1 does not contain the SOTA performance models on these datasets. Also, as I have said, I will not call this model completely "unsupervised."

**Summary Of The Paper:**

This paper presented a framework that combines self-supervised contrastive learning and neuro-symbolic method for "abstract visual reasoning," specifically grounded on the task of RAVEN's Progression Matrices (RPM). The main framework has four steps. First, it trains a per-object perception model that maps images to different embedding spaces such as shape, size, and color. Next, it uses self-supervised learning objectives to learn "possible rules" from the images, leveraging the definition of RPM matrices. In the third step, the model maps tokens into "symbolic" tokens (in particular natural numbers) based on an given set of possible rules in RPM. Finally, the model finetunes the model using "pseudo-labeling" for symbols.

**Summary Of The Review:**

See my weakness discussions for details.

---

### Official Review · Reviewer_FVX7 · 2022-10-24

**Confidence:** 4
**Correctness:** 2
**Technical Novelty And Significance:** 3
**Empirical Novelty And Significance:** 2
**Recommendation:** 3

**Clarity, Quality, Novelty And Reproducibility:**

The paper is mostly clearly written. More detail could be added to the description of the 'symbolic mapping' component (section 3.3). For instance, that section states 'we train a MLP to map the neural representations to the symbols used for reasoning the rules', but it is not explained how the MLP does this or how it is trained.

**Strength And Weaknesses:**

# Strengths:
- The proposed method achieves strong results on the RAVEN dataset, a popular benchmark for evaluating abstract visual reasoning abilities.
- The authors also evaluate the proposed method on the V-PROM dataset (with favorable results), which extends the RPM problem format to include naturalistic images. This dataset is a welcome introduction to the visual reasoning literature (usually focused on problems composed of abstract geometric forms), and it is good to see the authors adopt it.

# Weaknesses:
- The primary weakness is that the proposed method, NS-SSCL, which the authors characterize as 'self-supervised', is arguably not actually self-supervised. In fact, the method involves stronger supervision than some of the other methods that are characterized as 'supervised'. In particular, the contrastive learning used to shape perceptual representations in NS-SSCL depends on having groundtruth annotations of the perceptual attributes (e.g. object size, shape, etc.), whereas in many 'supervised' methods (which are supervised at the level of answer choice selection) these perceptual representations are learned end-to-end in the service of the downstream reasoning task. The paper characterizes the training procedure for perceptual encoding as 'self-supervised' because a contrastive loss is employed, similar to the SimCLR method of Chen et al. (2020). However, there is a crucial difference between SimCLR and the contrastive loss employed in the present work. In SimCLR, certain image transformations are employed (e.g. rotation, color distortion, etc.), and the model is trained to generate embeddings that are similar for both an original image and a transformed version of that image. The major advantage of this method is that one does not need image labels at all during training, thus it is *self*-supervised. In NS-SSCL, by contrast, the same kind of contrastive loss is used, however the positive and negative pairs used to compute the loss are defined in terms of groundtruth attributes. For instance, a positive pair for the 'size' attribute might consist of two objects of the same size (but potentially of different shapes, colors, etc.). This is very different than applying generic image transformations, since it actually requires the attribute labels in order to define what constitutes a positive example. Thus, even though the model is not trained explicitly to label the attributes, computing the loss requires the exact same level of supervision as if the model had been trained to generate labels. So, although it is correct to refer to this as a contrastive loss, it is not correct to refer to this as self-supervised learning, since supervision is required.
    - Note that I am not arguing that supervised learning of perceptual attributes is necessarily unreasonable. I think a reasonable argument can be made that, at least for human reasoners, perceptual representations are not learned in the context of learning to perform visual reasoning tasks, and so can be treated as the product of a separate learning process. However, it is very misleading to call this 'self-supervised' learning, and to contrast it with other 'supervised' methods, which actually require weaker supervision. It is also unclear, if the method requires groundtruth perceptual annotations, what advantage is conferred by using a contrastive approach as opposed to a more straightforward supervised classification method.
- I had some difficulty understanding how the symbolic reasoning module works, but if I understand correctly, it assumes prior knowledge of the rules employed in these problems, and also employs supervised training to map the perceptual representations (learned through contrastive learning) to the relevant symbols in the rule representations. This is an even stronger form of supervision, and makes it so that the model cannot process any rules other than those built into it by the modeler.
- The paper focuses on the RAVEN benchmark, but does not evaluate the method on PGM, the other major benchmark of RPM-like problems. This is presumably because PGM, unlike RAVEN, does not contain groundtruth annotations for the perceptual attributes, and so it is not possible to apply NS-SSCL to that benchmark, underscoring the reliance of the model on such strong supervision.

## Other comments:
- It seems that the method requires groundtruth object segmentations, is this the case? This aspect of the model is not addressed in the paper.
- The axes for figures 3 and 4 are extremely small and almost impossible to read.

**Summary Of The Paper:**

This paper proposes a novel method for solving visual reasoning (specifically RPM-like) problems, that combines contrastive losses at both the perceptual and abstract levels, together with a symbolic rule induction module. The method achieves high accuracy on both the RAVEN and V-PROM datasets.

**Summary Of The Review:**

The proposed method achieves strong results on the RAVEN benchmark, but the model is misleadingly characterized as 'self-supervised' when in fact it requires stronger supervision than many other models. The method is not evaluated on PGM, presumably because of the difficulty imposed by reliance on such strong supervision.

---

### Decision · Program_Chairs · 2023-01-20

**Decision:**

Reject

**Justification For Why Not Higher Score:**

1) Grave doubts about whether the approach should be considered unsupervised, and hence whether claims of superior performance are correct.

2) Proposed method leverages meta-information specific to the RPM task, unlike some other methods, severely limiting its generality and usefulness as an abstract reasoning method.

(Note: the authors did not submit any responses to the reviews)

**Justification For Why Not Lower Score:**

n/a

**Metareview: Summary, Strengths And Weaknesses:**

This paper proposes a model called NS-SSCL (neuro-symbolic self-supervised contrastive learning) for solving abstract (visual) reasoning problems. The concrete problems tackled are RPM-like (Raven's Progressive Matrices) tasks, with the specific datasets used being RAVEN and V-PROM. The overall pipeline is as follows. First, a perception module learns visual representations disentangled into different spaces for attributes such as shape, color and size. Second, a reasoning module learns possible rules in a self-supervised manner. Then, a mapping from representations into symbols/tokens is learnt. In terms of results, NS-SSCL purportedly achieves performance comparable to some SOTA supervised baselines.


--STRENGTHS--

The overall premise of the paper is novel and timely on multiple fronts. These include the aim of tackling RPM-like tasks with reduced supervision, and relatedly, the use of self-supervised methods on these tasks.

The specific chosen approach of combining neuro-symbolic methods with self-supervised learning seems to be an interesting and novel one.

In terms of empirical evaluation, the proposed method was tested on standard benchmarks such as RAVEN and V-PROM, with purportedly favorable results. (Although reviewers noted that benchmarking on another common dataset, PGM, was not done)


--WEAKNESSES--

Unanimously, there were grave doubts as to whether the proposed method should really be considered as purely/fully self-supervised or unsupervised. Furthermore, as a result of these doubts, there was further doubt as to whether the result comparisons were fair or like-for-like comparisons.

The basis for these doubts are as follows:
1) The learning of shape representations depends on access to ground-truth information about perceptual attributes such as color and shape, etc.
2) Although constrastive loss (from self-supervised learning approaches) is employed, the positive and negative pairs used here are in terms of the above ground-truth attributes.
3) As such, if the proposed method is taken to be a supervised one instead, then there is no convincing improvement over previous methods.

To be fair, the authors did somewhat allude to the above limitation in their concluding paragraph, saying that the model could be pre-trained on independent tasks first, to develop general visual and cognitive abilities. While that may or may not ultimately address this issue satisfactorily, for this paper in its current form, it is clear that there are sufficient serious doubts that I cannot recommend acceptance currently.

In addition to the above fatal issue, the proposed method also leverages information or assumptions specific to the RPM task, such as division into discrete cells, as well as knowledge about the structure of the rules governing the task. These severely limit the generality of the proposed method to be highly RPM-specific, rather than to be about abstract reasoning more generally.

**Summary Of Ac-Reviewer Meeting:**

n/a